Prevalence of depression and its associated factors among adolescents in China during the early stage of the COVID-19 outbreak

Qi Han 1
Liu Rui 1
Feng Yuan 1
Luo Jia 1
http://orcid.org/0000-0002-0957-5922 Lei Si Man 2
Cheung Teris 3
Ungvari Gabor S. 4 5
Chen Xu 1 chenxuadyy@ccmu.edu.cn
Xiang Yu-Tao 6 7 ytxiang@um.edu.mo
1 The National Clinical Research Center for Mental Disorders & Beijing Key Laboratory of Mental Disorders, Beijing Anding Hospital & the Advanced Innovation Center for Human Brain Protection, Capital Medical University , Beijing , China
2 Faculty of Education, University of Macau , Macao SAR , China
3 School of Nursing, Hong Kong Polytechnic University , Hong Kong SAR , China
4 Division of Psychiatry, School of Medicine, University of Western Australia , Perth , Australia
5 The University of Notre Dame Australia , Fremantle , Australia
6 Unit of Psychiatry, Department of Public Health and Medicinal Administration, & Institute of Translational Medicine, Faculty of Health Sciences, University of Macau , Macao SAR , China
7 Centre for Cognitive and Brain Sciences, University of Macau , Macao SAR , China
Zhong Bao-Liang
Electronic publication date: 2021 Nov 11
Publication date: 2021
Volume: 9
Electronic Location ID: e12223
Received 2021 Feb 24; Accepted 2021 Sep 7
Copyright: © 2021 Qi et al.
Copyright year: 2021
Copyright holder: Qi et al.
License: This is an open access article distributed under the terms of the Creative Commons Attribution License, which permits unrestricted use, distribution, reproduction and adaptation in any medium and for any purpose provided that it is properly attributed. For attribution, the original author(s), title, publication source (PeerJ) and either DOI or URL of the article must be cited.
License URL: https://creativecommons.org/licenses/by/4.0/

Keywords: Adolescents, China, COVID-19, Depression, Prevalence

Funding: National Science and Technology Major Project for investigational new drug 2018ZX09201-014 The Beijing Municipal Science & Technology Commission Z181100001518005; Z181100001718124 National Key R&D Program of China 2017YFC1311100 Beijing Municipal Science & Tech Commission D171100007017001 University of Macau MYRG2019-00066-FHS The study was supported by the National Science and Technology Major Project for investigational new drug (2018ZX09201-014), the Beijing Municipal Science & Technology Commission (Z181100001518005; Z181100001718124), the National Key R&D Program of China (2017YFC1311100), the Beijing Municipal Science & Tech Commission (D171100007017001), and the University of Macau (MYRG2019-00066-FHS). The funders had no role in study design, data collection and analysis, decision to publish, or preparation of the manuscript.

==============================
Background

The outbreak of the 2019 coronavirus disease outbreak (COVID-19) had a detrimental impact on adolescents’ daily life and studying, which could increase the risk of depression. This study examined the prevalence of depressive symptoms (depression hereafter) among Chinese adolescents and its associated factors.

Methods

An online survey was conducted during the COVID-19 outbreak in China. Adolescents aged 11–20 years who currently lived in China were invited to participate in the study. Data were collected with the “SurveyStar” platform using the Snowball Sampling method. Depression was assessed using the Center for Epidemiologic Studies Depression Scale (CES-D).

Results

A total of 9,554 adolescents participated in the study. The prevalence of depression was 36.6% (95% CI [35.6%–37.6%]); the prevalence of mild and moderate-severe depression was 9.2% (95% CI [8.9%–9.5%]) and 27.4% (95% CI [26.9%–27.9%]), respectively. Female gender (OR = 1.235, P < 0.001), senior secondary school grade (OR = 1.513, P < 0.001), sleep duration of <6 h/day (OR = 2.455, P < 0.001), and living in Hubei province (total number of infected cases > 10,000) (OR = 1.245, P = 0.038) were significantly associated with higher risk of depression. Concern about COVID-19 (OR = 0.632, P < 0.001), participating in distance learning (OR = 0.728, P = 0.001), sleep duration of >8 h/day (OR = 0.726, P < 0.001), exercise of >30 min/day, and study duration of ≥4 h/day (OR = 0.835, P < 0.001) were associated with lower risk of depression.

Conclusion

Depression was common among adolescents in China during the COVID-19 outbreak. Considering the negative impact of depression on daily life and health outcomes, timely screening and appropriate interventions are urgently needed for depressed adolescents during the COVID-19 outbreak.

Introduction

In December 2019, the coronavirus disease 2019 (COVID-19) outbreak was first reported in China, and then was found in more than 100 countries. On the 11th of March 2020, the World Health Organization (WHO) declared COVID-19 as a pandemic (World Health Organization, 2020).

With the rapid transmission of the COVID-19, mental health problems have been more common in different segments of the population, such as confirmed/suspected cases, frontline medical professionals, and the elderly (Xiang et al., 2020; Yang et al., 2021; Yang et al., 2020). However, little is known about mental health of adolescents, although this is arguably a vulnerable subpopulation to experience mental health problems, particularly depression due to academic pressure (Li & Zhang, 2014), negative life events (Li, 2016), and low self-esteem (Lian et al., 2016). To reduce the rapid transmission, mass quarantine has been adopted and all face-to face teaching in schools has been suspended in China during the COVID-19 outbreak. Distance learning and online teaching have been adopted for adolescents nationwide. Considering the uncertainty, fear, boredom, loneliness and anger associated with quarantine, challenges brought up by the sudden changes of traditional learning methods coupled with academic pressures, conflicts with parents, reduced social interaction with peers, and lack of outdoor activities, the risk of depressive symptoms (depression hereafter) was expected to increase among adolescents during the COVID-19 outbreak.

Several surveys on the prevalence of depression in adolescents have been conducted in China during the COVID-19 outbreak yielding conflicting findings: prevalence figures ranged from 6.41% (Tang & Pang, 2020), 10.4% (Wang et al., 2020c), 11.88% (Chen et al., 2020), 21.16% (Chang, Yuan & Wang, 2020) to 43.7% (Zhou et al., 2020). Commonly reported correlates of depression during the outbreak included female gender, high school grade, decreased frequency of physical exercise, overuse of the internet and social media and familial vulnerability (Guessoum et al., 2020; Kilincel et al., 2021). The discrepancy of the findings between studies could be partly due to different sampling methods, rating instruments (e.g., the Patient Health Questionnaire (PHQ-9), the Depression Self-Rating Scale for Children (DSRS-C), and the Children’s Depression Inventory (CDI)) and studies conducted at different stages of the COVID-19 outbreak. Many of these studies were completed after the peak of the COVID-19 outbreak (i.e., after February 2020), therefore their findings could not reflect the true spectrum of adolescents’ mental health at the beginning of the outbreak (Chen et al., 2020; Zhou et al., 2020). Furthermore, nationwide data at the early stage of the outbreak were rarely reported (Tang & Pang, 2020; Wang et al., 2020c). To bridge the gap in knowledge of the impact of COVID-19 on adolescents, an online survey was conducted to investigate the prevalence of depression and explore its associated factors in a large cohort of secondary school students in China in the early stage of COVID-19 outbreak.

Materials & methods

Study design and participants

This was an online survey conducted between February 20th and 27th, 2020 within the context of the collaborative research network of the National Clinical Research Center for Mental Disorders, China (Qi et al., 2020). Data were collected with the WeChat-based “SurveyStar” program (https://www.wjx.cn/). Snowball sampling was used. The WeChat program has been widely used on student management in most secondary schools in China. To be eligible, participants should be: (1) secondary school students aged between 11 and 20 years; and (2) currently living in mainland China during the COVID-19 outbreak. Participants with severe mental disorders based on health records at schools or those who refused to participate were excluded in this study. All participants were required to provide electronic written informed consent. For those under 18 years old, participants’ legal guardians were also required to provide electronic written consent. This study was approved by the Medical Ethical Committee in Beijing Anding Hospital of the Capital Medical University, China ((2020)KEYAN(No.10)).

Assessment instruments and data collection

A data sheet designed for this study was used to collect socio-demographic and clinical characteristics, such as gender, age, place of residence, school grade, parents’ occupations (frontline workers or not), attitude towards COVID-19, duration of physical exercise, attendance at distance learning, and presence of depression. Study sites were classified according to the total number of COVID-19 patients at provincial level based on the report of the National health commission of China (http://www.nhc.gov.cn) released on the February 27th, 2020.

The presence of depression was assessed by the Chinese version of the Center for Epidemiological Studies Depression Scale (CES-D) (Radloff, 1977). The CES-D is a 20-item self-reported questionnaire with satisfactory psychometric properties in Chinese adolescents (Chen, Yang & Li, 2009). Each item is scored from 0 (not at all) to 3 (a lot). Participants with a CES-D total score of >15 was considered as ‘depressed’ (Zhang et al., 2010); those with a CES-D total score of 16–19 were considered ‘mildly depressed’, and a CES-D total score of >19 indicated ‘moderate to severe depression’ (Gao, 2009).

Statistical analyses

SPSS 21.0 was used to analyze data. Chi-square tests were used to compare demographic characteristics between depression and ‘non-depression’ groups. Multivariate logistic regression analyses with the ‘enter’ method was performed to examine independent correlates of depression, with depression as the dependent variable, and those having significant group differences in univariate analyses as independent variables. The level of significance was 0.05 (two-tailed).

Results

With the exception of Tibet, all provinces of mainland China participated in this study. A total of 9,744 adolescents were invited to participate, of whom, 9,554 met the study criteria and were included in the analyses. The overall prevalence of depression was 36.6% (95% CI [35.6%–37.6%]), with the prevalence of mild and moderate-severe of 9.2% (95% CI [8.9%–9.5%]) and 27.4% (95% CI [26.9%–27.9%]), respectively.

Table 1 shows the socio-demographic and clinical characteristics comparing the depression and ‘non-depression’ groups. Univariate analyses revealed that age, gender, place of residence, school grade, infected acquaintances, attitudes toward COVID-19, sleep duration, duration of daily physical exercise, study time, and distant learning were significantly associated with depression (P < 0.05). Figures 1 and 2 presents the CES-D total scores according to school grades and duration of physical exercise.

Figure 1 The CES-D mean scores and standard errors by different grades of adolescents during the COVID-19 outbreak.

Figure 2 The CES-D mean scores and standard errors by physical activities of adolescents during the COVID-19 outbreak.

Table 1 Demographic characteristics of the study sample (N = 9,554).

Variables	Total	Depression	Non-depression	X 2	P-value	
(N = 9,554)	(N = 3,498)	(N = 6,056)	
n	%	n	%	n	%	
Age (years)							176.33	<0.001	
11–15	5,607	58.7	1,745	49.9	3,862	63.8			
16–20	3.947	41.3	1,753	50.1	2,194	36.2			
Male gender	4,577	47.9	1,514	43.3	3,063	50.6	47.30	<0.001	
Having infected acquaintances	336	3.5	143	4.1	193	3.2	5.31	0.021	
Parents as frontline workers	823	8.6	312	8.9	511	8.4	0.65	0.419	
Be concerned about COVID-19	7,639	80.0	2,580	73.8	5,059	83.5	132.35	<0.001	
Study duration ≥ 4 h/day	8,012	83.9	2,794	79.9	5,218	86.2	64.78	<0.001	
Distance learning	8,953	93.7	3,181	90.9	5,772	95.3	71.92	<0.001	
Grade							172.10	<0.001	
Junior secondary school	5,459	57.1	1,693	48.4	3,766	62.2			
Senior secondary school	4,095	42.9	1,805	51.6	2,290	37.8			
Living area (No. of infected patients)							8.72	0.033	
10–99	2,224	23.3	825	23.6	1,399	23.1	–	–	
100–999	5,177	54.2	1,932	55.2	3,245	53.6	–	–	
1,000–9,999	1,631	17.1	545	15.6	1,086	17.9	–	–	
>10,000	522	5.5	196	5.6	326	5.4			
Sleep duration/day							221.30	<0.001	
<6 h	446	4.7	283	8.1	163	2.7	–	–	
6–8 h	5,359	56.1	2,089	59.7	3,270	54.0	–	–	
>8 h	3,749	39.2	1,126	32.2	2,623	43.3			
Physical exercise duration/day (indoor and outdoor)							181.55	<0.001	
<30 min	4,392	46	1,924	55	2,468	40.8	–	-	
30–60 min	4,250	44.5	1,289	36.8	2,961	48.9	–	-	
>60 min	912	9.5	285	8.1	627	10.4			
Residence							2.05	0.152	
Dorm	74	0.8	33	0.9	41	0.7	–	–	
Others	9,480	99.2	3,465	99.1	6,015	99.3			
Note:

Provinces were classified according to the number of confirmed cases of COVID-19 for where the students lived; Depression was defined as total score of the Center for Epidemiological Studies Depression Scale for Children (CES-D) > 15; COVID-19: Coronavirus disease 2019; Bolded values: P < 0.05.

Table 2 shows the results of multivariate logistic regression analyses. Females (OR = 1.235, 95% CI [1.130–1.348], P < 0.001), and senior secondary school students (OR = 1.513, 95% CI [1.379–1.661], P < 0.001), sleep duration < 6 h/day (OR = 2.455, 95% CI [1.998–3.016], P < 0.001), and living in Hubei province (total number of infected cases > 10,000) (OR = 1.245, 95% CI [1.013–1.530], P = 0.038) were significantly associated with higher risk of depression. In contrast, students who concerned about the COVID-19 (OR = 0.632, 95% CI [0.568–0.703], P < 0.001), sleep duration > 8 h/day (OR = 0.726, 95% CI [0.662–0.796], P < 0.001), participating in distance learning (OR = 0.728, 95% CI [0.608–0.873], P = 0.001), duration of physical exercise > 30 min/day, and duration of study ≥ 4 h/day (OR = 0.835, 95% CI [0.740–0.943], P < 0.001) were significantly associated with lower risk of depression.

Table 2 Independent correlates of depression by multivariate logistical regression analysis.

Variables	P-value	OR	95% CIs for OR	
Lower	Upper	
Female	<0.001	1.235	1.130	1.348	
Senior secondary school	<0.001	1.513	1.379	1.661	
Having infected acquaintances	0.063	1.241	0.988	1.559	
Concerned about COVID-19	<0.001	0.632	0.568	0.703	
Sleep duration/day					
6–8 h	ref	–	–	–	
<6 h	<0.001	2.455	1.998	3.016	
>8 h	<0.001	0.726	0.662	0.796	
Physical exercise duration/day (indoor and outdoor)					
<30 min	ref	–	–	–	
30–60 min	<0.001	0.636	0.580	0.698	
>60 min	<0.001	0.686	0.586	0.803	
Study duration/day					
<4 h	ref	–	–	–	
≥4 h	0.004	0.835	0.740	0.943	
Distance learning	0.001	0.728	0.608	0.873	
Living area (No. of infected patients)					
10–99	ref	–	–	–	
100–999	0.236	1.068	0.958	1.191	
1,000–9,999	0.002	0.797	0.691	0.918	
>10,000	0.038	1.245	1.013	1.530	
Note:

Due to collinearity between age and grade, age group was not entered in the multiple logistic regression analysis. Bolded values: P < 0.05; CI, confidential interval; OR, odds ratio.

Discussion

To the best of our knowledge, this was the first survey examining the prevalence of depression and its associated factors among adolescents during the COVID-19 outbreak. The main finding of the survey is that 36.6% (95% CI [35.6%–37.6%]) of Chinese adolescents suffered from depression as defined by a score of >15 on the self-reported CES-D; of whom 27.4% presented with moderate to severe depression (95% CI [26.9%–27.9%]). In China, there were 148.4 million adolescents in 2018 (National Bureau of Statistics of China, UNICEF China, & UNFPA China, 2018), suggesting that approximately 54.3 million adolescents could be suffering from broadly defined depression extrapolating the results of this study.

A recent meta-analysis found that the pooled prevalence of depression in secondary school students in mainland China was 24.3% (95% CI [21.3%–27.6%]) (Tang et al., 2019). Compared to Tang et al. (2019) findings, the current study figures were higher, indicating that the COVID-19 outbreak was a possibly significant risk factor contributing to the development of depression among adolescents. The anxiety-provoking experience brought about by the COVID-19 outbreak, such as mass quarantine, limited public transportation, fear of infection, frustration, boredom, anger, lack of interpersonal contact with peers, limited personal space at home, financial loss in the family, uncertainty about the future, were highly likely associated with the growing risk of depression (Brooks et al., 2020; Wang et al., 2020b; Wang et al., 2016).

Consistent with previous studies (Leung et al., 2020; Zhang et al., 2020; Zhou et al., 2020), female students were more likely to suffer from depression. The gender difference in the prevalence of depression could be partly attributed to endocrine and neurodevelopmental during the pubertal transition (Hyde, Mezulis & Abramson, 2008) and more frequent introvert and vulnerable personality traits in girls (Salk, Hyde & Abramson, 2017). During the COVID-19 outbreak, all primary and secondary schools in China were temporally closed to reduce the rapid COVID-19 transmission. Students were instructed to stay at home, reduce visits to others and limit outdoor activities. Nevertheless, increased physical exercise, sleep and study duration were associated with lower risk of depression. Other studies confirmed that physical exercise (Carter et al., 2016), and sleeping up to 8–9 h/day (Chiu et al., 2018; Guo et al., 2017) could reduce the likelihood of depression in adolescents. The present study also confirmed previous findings (Guo et al., 2017; Raniti et al., 2017) that sleeping for less than 8 h was associated with increased risk of depression.

Distance learning was associated with decreased risk of depression, which may be attributed to regular study schedules under teachers’ supervision (Guo et al., 2017). Senior secondary school students had higher risk of depression than junior students. Compared to other countries, secondary school students in China may have higher study pressure from schools and parents, which may exacerbate the risk of depression (Tang et al., 2020). A recent study suggested that Chinese educational authorities should issue guidelines on effective online learning, provide online education on healthy lifestyle and online psychosocial support programmes (Wang et al., 2020b) to relieve the study pressure on adolescents.

Adolescents living in Hubei province had a higher risk of depression than other areas of China. Hubei province was probably the original epicenter of the COVID-19 outbreak in China. Therefore, a wide range of strict quarantine measures were immediately adopted to prevent rapid transmission of infection to other provinces. Prolonged quarantine has negative impact on mental health (Brooks et al., 2020), which could explain the higher risk of depression in the current study. Additionally, the large number of infected cases and deaths in Hubei province undoubtedly created an atmosphere of intense fear in the community elevating the risk of mental health problems, particularly depression among the residents.

Students who were more concerned about the COVID-19 had lower risk of depression, indicating that clear communication, regular and accurate updates about the COVID-19 reduced uncertainty and fear, and consequently the likelihood of depression (Xiang et al., 2020). Since the COVID-19 outbreak, daily press release by health authorities have disseminated updated information about the epidemic in many regions of China (Wang et al., 2020a). The National Health Commission of China also released national guidelines of psychological crisis intervention related to the COVID-19 (Kang et al., 2020). Consequently, students’ knowledge and better understanding about the COVID-19 epidemic were likely to substantially reduce their uncertainty and fear (Li & Reavley, 2021).

The strengths of this study included the large sample size covering almost all major areas of China. However, several limitations needed to be addressed. First, the design of our online survey meant that an unknown number of students without access to the internet could not participate in this study. However, the likelihood of not having internet access must have been very low, because of the near universal delivery of online teaching in schools in China. Second, relevant factors associated with depression, such as peer social support, economic status, academic performance, and relationships with families, could not be examined due to logistical reasons. Lastly, causality between variables could not be examined due to the cross-sectional design of this study.

Conclusions

Depression was common among adolescents during the COVID-19 pandemic. Considering the negative impact of depression on activities of daily life and health outcomes, timely screening and appropriate interventions, such as online psychological counselling, are urgently needed to reduce the likelihood of depression among adolescents during the current COVID-19 epidemic as well as in similar possibly future infectious disease outbreaks.

Supplemental Information

Supplemental Information 1 Raw data.

Click here for additional data file.

Supplemental Information 2 Codebook for the raw data.

Click here for additional data file.

Supplemental Information 3 Questionnaire for the current study.

Click here for additional data file.

Additional Information and Declarations

Competing Interests

Author Contributions

Human Ethics

Data Availability

The authors declare that they have no competing interests.

Han Qi conceived and designed the experiments, performed the experiments, analyzed the data, prepared figures and/or tables, authored or reviewed drafts of the paper, and approved the final draft.

Rui Liu conceived and designed the experiments, performed the experiments, analyzed the data, prepared figures and/or tables, authored or reviewed drafts of the paper, and approved the final draft.

Yuan Feng performed the experiments, prepared figures and/or tables, and approved the final draft.

Jia Luo performed the experiments, authored or reviewed drafts of the paper, and approved the final draft.

Si Man Lei performed the experiments, authored or reviewed drafts of the paper, and approved the final draft.

Teris Cheung performed the experiments, authored or reviewed drafts of the paper, and approved the final draft.

Gabor S. Ungvari performed the experiments, authored or reviewed drafts of the paper, and approved the final draft.

Xu Chen conceived and designed the experiments, performed the experiments, authored or reviewed drafts of the paper, and approved the final draft.

Yu-Tao Xiang conceived and designed the experiments, authored or reviewed drafts of the paper, and approved the final draft.

The following information was supplied relating to ethical approvals (i.e., approving body and any reference numbers):

Beijing Anding Hospital of the Capital Medical University.

The following information was supplied regarding data availability:

The raw data are available in the Supplemental File.

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
