# Peer review of "Prevalence of depression and its associated factors among adolescents in China during the early stage of the COVID-19 outbreak"

_PeerJ, doi:10.7717/peerj.12223_

## Round 0.1 · original submission · Major Revisions

Please consider the reviewers' comments and revise the paper accordingly.

Reviewer 1 ·

Basic reporting

This was a cross-sectional online study. It investigated the prevalence of depression among Chinese adolescents during COVID-19 and its associated factors. It found the prevalece of depression was 36.6%. Female, senior secondary school students, sleep duration <6 hours/day, and living in Hubei province were significantly associated with higher risk of depression.It was written well in English.

Experimental design

The research question was well defined.Methods were described with sufficient information.

Validity of the findings

The study found depression was common among adolescent during the COVID-19 break and interventions are needed for depressed adolecents among the diesease pandemic. Conclusion are well stated.

Additional comments

Materials & Methods
Line 86 Participant with severe mental disorder were excluded. It was easy to describe. However, how did you exclude these participants during investigation? This may be not an exclusion.

Discussion
Female was associated with higher risk of depression. Please discuss the possible reason.

Reviewer 2 ·

Basic reporting

1. In the methods part of abstract, please briefly describe the inclusion of subjects and data collected during the online survey. In the results part of abstract, please provide statistics such as OR and P values of factors associated with depressive symptoms.

2. Please carefully check the English language of this manuscript such as “negative detrimental impact” in line 36-37.

3. The sample is not nationally representative, so the authors may consider to tone down the title “national survey”.

4. Introduction. The authors focused on 1. Prevalence of and 2. Factors associate with depression in adolescents during the pandemic, so a brief review on existing studies on the two aspects of adolescents during the pandemic is necessary.

5. Psychosocial, academic, and family related factors are commonly reported as correlates of depressive symptoms in adolescents. Candidate factors included in this study are very limited.

6. It would be interesting to report the prevalence of depressive symptoms of varying severities according to CESD scores.

Experimental design

1. In the methods part of abstract, please briefly describe the inclusion of subjects and data collected during the online survey. In the results part of abstract, please provide statistics such as OR and P values of factors associated with depressive symptoms.

2. Please carefully check the English language of this manuscript such as “negative detrimental impact” in line 36-37.

3. The sample is not nationally representative, so the authors may consider to tone down the title “national survey”.

4. Introduction. The authors focused on 1. Prevalence of and 2. Factors associate with depression in adolescents during the pandemic, so a brief review on existing studies on the two aspects of adolescents during the pandemic is necessary.

5. Psychosocial, academic, and family related factors are commonly reported as correlates of depressive symptoms in adolescents. Candidate factors included in this study are very limited.

6. It would be interesting to report the prevalence of depressive symptoms of varying severities according to CESD scores.

Validity of the findings

1. In the methods part of abstract, please briefly describe the inclusion of subjects and data collected during the online survey. In the results part of abstract, please provide statistics such as OR and P values of factors associated with depressive symptoms.

2. Please carefully check the English language of this manuscript such as “negative detrimental impact” in line 36-37.

3. The sample is not nationally representative, so the authors may consider to tone down the title “national survey”.

4. Introduction. The authors focused on 1. Prevalence of and 2. Factors associate with depression in adolescents during the pandemic, so a brief review on existing studies on the two aspects of adolescents during the pandemic is necessary.

5. Psychosocial, academic, and family related factors are commonly reported as correlates of depressive symptoms in adolescents. Candidate factors included in this study are very limited.

6. It would be interesting to report the prevalence of depressive symptoms of varying severities according to CESD scores.

Additional comments

1. In the methods part of abstract, please briefly describe the inclusion of subjects and data collected during the online survey. In the results part of abstract, please provide statistics such as OR and P values of factors associated with depressive symptoms.

2. Please carefully check the English language of this manuscript such as “negative detrimental impact” in line 36-37.

3. The sample is not nationally representative, so the authors may consider to tone down the title “national survey”.

4. Introduction. The authors focused on 1. Prevalence of and 2. Factors associate with depression in adolescents during the pandemic, so a brief review on existing studies on the two aspects of adolescents during the pandemic is necessary.

5. Psychosocial, academic, and family related factors are commonly reported as correlates of depressive symptoms in adolescents. Candidate factors included in this study are very limited.

6. It would be interesting to report the prevalence of depressive symptoms of varying severities according to CESD scores.

---

## Round 0.2 · accepted · Accept

I am pleased to accept this interesting paper.

Reviewer 1 ·

Basic reporting

The article included sufficient introduction and background. It was well written in English.

Experimental design

The submission clearly define the research question.

Validity of the findings

The conclusions was appropriately stated

Additional comments

none

Reviewer 2 ·

Basic reporting

I have no further comments. The authors have addressed my concerns.

Experimental design

I have no further comments. The authors have addressed my concerns.

Validity of the findings

I have no further comments. The authors have addressed my concerns.

Additional comments

I have no further comments. The authors have addressed my concerns.